# Natural Cellulose-Based Multifunctional Nanofibers for the Effective Removal of Particulate Matter and Volatile Organic Compounds

**DOI:** 10.3390/nano13111720

**Published:** 2023-05-24

**Authors:** Sang Hyun Ji, Ji Sun Yun

**Affiliations:** New Growth Materials Division, Korea Institute of Ceramic Engineering and Technology, 101, Soho-ro, Jinju 52851, Republic of Korea; sanghyun_ji@kicet.re.kr

**Keywords:** cellulose, nanofiber, ionic liquid, air filter, toluene adsorption

## Abstract

Multifunctional nanofibers for particulate matter (PM) and volatile organic compounds (VOCs) removal from the indoor atmospheric environment were manufactured from eco-friendly natural cellulose materials via electrospinning using an optimized solvent system containing 1-ethyl-3-methylimidazolium acetate (EmimAC) and dimethylformide (DMF) in a 3:7 volume ratio. EmimAC improved the cellulose stability, whereas DMF improved the electrospinnability of the material. Various cellulose nanofibers were manufactured using this mixed solvent system and characterized according to the cellulose type, such as hardwood pulp, softwood pulp, and cellulose powder, and cellulose content ranging from 6.0–6.5 wt%. The correlation between the precursor solution alignment and electrospinning properties indicated an optimal cellulose content of 6.3 wt% for all cellulose types. The hardwood pulp-based nanofibers possessed the highest specific surface area and exhibited high efficiency for eliminating both PM and VOCs, with a PM_2.5_ adsorption efficiency of 97.38%, PM_2.5_ quality factor of 0.28, and toluene adsorption of 18.4 mg/g. This study will contribute to the development of next-generation eco-friendly multifunctional air filters for indoor clean-air environments.

## 1. Introduction

As people spend more than 90% of their time indoors owing to the COVID-19 pandemic [1], the concentrations of indoor pollutants such as particulate matter (PM) and volatile organic compounds (VOCs) have increased to two to five times those observed outdoors [2]. Outgassing and particulate emissions from adhesives, paints, and building materials are the primary sources of indoor air pollutants. Prolonged exposure to these pollutants may impair the respiratory and central nervous systems [3,4]. Therefore, methods to improve indoor air quality through effective and efficient removal of PM and VOCs are required.

The principles to remove PM and VOCs are different [5,6]: PM can be removed by sieving, gravity settling, inertial impaction, interception, diffusion, and electrostatic attraction, whereas VOCs can be adsorbed by highly functional filters having a large specific surface area, porous structure, and high surface reactivity. Simultaneous removal of PM and VOCs is typically accomplished using multilayered filter modules, or filter media composited with adsorbents such as activated carbon and zeolite [7]. However, the structures of these two filters tend to be bulky and heavy, limiting their application. Therefore, various thin and light filters have recently been developed, including the hierarchical-structured porous hybrid filter [8] and the polylactic acid-cyclodextrins-based fiber filter [9]. However, because these filters comprise synthetic polymers, secondary pollutants are generated after use, and compact and slim air filters constructed from eco-friendly natural materials are in demand.

Cellulose-based nanofibers can serve as an alternative to synthetic materials for removing PM and VOCs owing to their mechanical properties, high porosity, chemical diversity, and natural biodegradability [10,11,12,13]. Cellulose-based nanofibers can be manufactured using various methods, including electrospinning, phase separation, and plasma induction. The electrospinning process is best suited for manufacturing high-efficiency air filters because it produces uniform nanofibers with a high specific surface area and excellent internal connectivity [14,15]. However, very few solvents adequately dissolve cellulose to prepare precursor solutions for electrospinning. Solvents such as H_2_SO_4_ and trifluoroacetic acid are hazardous to the human body [16,17], whereas the insufficient volatility of other solvents hinders the electrospinning process. Therefore, non-toxic mixed solvent systems are required to enhance cellulose solubility and electrospinnability simultaneously.

In this study, a mixed solvent precursor solution was developed, containing 1-ethyl-3-methylimidazolium acetate (EmimAC) to dissolve the cellulose in an eco-friendly manner and dimethylformide (DMF) to improve the electrospinning stability. The EmimAC-DMF mixed solvent was then combined with various types of cellulose, including hardwood pulp (H-Pulp), softwood pulp (S-Pulp), and cellulose powder (Powder), to manufacture nanofibers. The removal of PM and VOCs using these cellulose nanofibers were investigated and characterized.

## 2. Material and Methods

The cellulose precursor solutions of H-Pulp (bleached hardwood kraft pulp, Sigma-Aldrich, St. Louis, MO, USA), S-Pulp (bleached softwood kraft pulp, Sigma-Aldrich, St. Louis, MO, USA), and cellulose powder (Sigma-Aldrich, St. Louis, USA) were prepared using a two-step process. Solutions of EmimAC (98.0%, Sigma-Aldrich, St. Louis, USA) and DMF (99.5%, Sigma-Aldrich, St. Louis, USA) were mixed in volume ratios of 5:5, 4:6, 3:7, 2:8, and 1:9, and stirred at 80 °C for 60 min. Subsequently, H-Pulp, S-Pulp, and Powder were added to the EmimAC-DMF solutions at concentrations of 6.0, 6.3, or 6.5 wt% and stirred at 80 °C for 60 min. The homogeneous cellulose precursor solution was then loaded into a 12 mL plastic syringe with a 20-gauge metal needle. The solution flow rate and the collector–tip distance was fixed at 1 mL h^–1^ and 15 cm, respectively, and the voltage between the needle and rotating drum collector were maintained between 15 to 20 kV. The as-spun cellulose nanofibers were stabilized at 80 °C for 24 h.

The cellulose precursor solution viscosities were measured using an LV-01 spindle and a viscometer (DV2TlV, Brookfield, MA, USA) at 6.0 rpm and 25 °C. The cellulose precursor solution solubilities were analyzed using solubility measurement equipment (NEPHELOster, BMG Labtrch, Ortenberg, Germany). The morphology of the cellulose precursor solution diluted by 100 times was analyzed using transmission electron microscopy (TEM; HT7700 microscope, Tokyo, Japan). The cellulose nanofiber morphologies were characterized using field-emission scanning electron microscopy (FE-SEM; JSM-7610F, JEOL, Akishima, Japan). Moreover, the specific surface area, pore size, and pore volume of the cellulose nanofibers were analyzed using a micropore physisorption analyzer (ASAP 2020MT, Micromeritics, Norcross, USA). The specific surface areas were determined using the Brunauer–Emmett–Teller (BET) equation. The pore size and volume of the micropores and mesopores were determined by the t-plot and Barrett–Joyner–Halenda (BJH) methods, respectively. The PM adsorption of the cellulose nanofibers was measured before and after filtration using a particle counter (EPAM-5000, SKC Inc., Covington, USA). PM with sizes of 0.5 μm (PM_0.5_), 1.0 μm (PM_1.0_), and 2.5 μm (PM_2.5_) was generated using a PM aerosol generator (QRJZFSQ-I, Beijing, China). The generated PM flowed into the cellulose nanofibers at a rate of 10 L/min for 8 h prior to the calculation of PM adsorption efficiency using Equation (1) [18]:(1)$${E\left(\%\right)=(C_{u}-C_{d})/C}_{u}\times 100$$
where *C_u_* and *C_d_* denote the PM concentration before and after filtration, respectively. The PM_2.5_ quality factor (*QF*) was calculated from the PM_2.5_ adsorption efficiency (*E*) and pressure drop (Δ*P*) using Equation (2) [19]:(2)QF=−ln(1−E)/∆P

The air permeability of the cellulose nanofibers was analyzed using a dynamic air permeability tester (FX 3350, TEXTEST AG, Schwerzenbach, Switzerland).

Toluene adsorption was determined using a catalytic reaction system (TENG Inc., Daejeon, Republic of Korea). Cellulose nanofibers (0.1 g) were loaded into a Pyrex glass reactor having an inner diameter of 1 cm, which was then installed within a catalytic reaction system and degassed for 3 h at 120 °C under argon gas. The adsorption isotherms of toluene (1000 ppm) were measured at 35 °C with a flow rate of 50 cm^3^/min, whereas the toluene concentration in the outlet flow was recorded using gas chromatography (GC-2010 Plus, SHIMAZU). The cellulose nanofibers were analyzed before and after toluene adsorption using an FT-IR spectrometer (Nicolet5700, Waltham, MA, USA) with a scanning range of 500 to 4000 cm^−1^.

## 3. Results and Discussion

Cellulose is insoluble in water and various other solvents owing to its high crystallinity and numerous hydrogen bonds within the cellulose fibers [20,21]. EmimAC is an eco-friendly ionic liquid that is effective in dissolving polysaccharides such as cellulose [22], which is mediated via the cleavage of hydrogen bonds between cellulose molecular chains (Figure 1a). However, it is challenging to regulate the viscosity of the EminAC-based precursor solution to make it acceptable for electrospinning. Furthermore, because EminAC does not undergo complete volatilization during electrospinning, the cellulose nanofibers are redissolved after electrospinning. Therefore, we prepared mixed solvent systems by mixing EminAC and DMF in various ratios to improve the cellulose solubility and electrospinning stability simultaneously. DMF is known to improve electrospinnability by decreasing the precursor solution viscosity while increasing its electrical conductivity [23]. The mixed solvents were then dissolved with various cellulose types (H-Pulp, S-Pulp, and Powder) to prepare the precursor solutions, designated as H-xPulp, S-xPulp, and xPowder, whereas the electrospun cellulose nanofibers were designated as H-xNF, S-xNF, and P-xNF (x indicates the cellulose content). The EminAC-DMF mixing ratio was first optimized for the H-6.3Pulp, S-6.3Pulp, and 6.3Powder samples according to viscosity and solubility (Figure 1b). The cellulose solubility sharply decreased in all precursor solutions, with DMF exceeding 70 vol%. The viscosity steadily decreased as the DMF was increased to 70 vol%, after which it remained constant or slightly increased with a further increase in DMF content owing to poor solubility. Typically, the precursor solution viscosity for electrospinning should range from 750 to 1000 cP [24]. Therefore, an optimized mixing ratio of 3:7 was determined for the EmimAC-DMF mixed solvent system.

The mixing periods of the precursor solutions with cellulose contents of 6.0, 6.3, and 6.5 wt% were also investigated (Figure 2a and Appendix A). The solubilities of all precursor solutions increased with mixing time, exhibiting over 99% solubilities after 60 min (Appendix A). The viscosity of each precursor solution initially increased and then decreased. The cellulose initially absorbed the solvent and swelled, thereby increasing the viscosity; thereafter, the cellulose dissolved in the solvent and organized itself anisotropically, which reduced the solution viscosity [25]. H-Pulp, which has the largest specific surface area (936 m^2^/g, Appendix A), absorbed the most solvent and exhibited the greatest swelling. The viscosity of its precursor solution rapidly increased in the first 30 min of mixing time and then sharply declined. For the Powder sample, which has the smallest specific surface area (716 m^2^/g), the viscosity of its precursor solution did not increase as much as the other two samples within the first 30 min. These samples exhibited solubilities greater than 99.4% and a viscosity range of 786–984 cP at a mixing time of 60 min (Appendix A). Therefore, 60 min was chosen as the optimal mixing time. 

The alignment of the precursor solutions and nanofibers containing pulp and powder cellulose samples differed. Well-aligned cellulose was observed in the TEM images (Figure 2c) of the pulp-type precursor solutions (H-6.3Pulp and S-6.3Pulp) but not in those of the powder-type precursor solution (6.3Powder). The electrospun nanofibers exhibited a similar behavior (Figure 2b,d), with the pulp-type H-6.3NF and S-6.3NF demonstrating stronger anisotropic effects than the powder-type P-6.3NF sample. Favorable anisotropic conditions in the precursor solution resulted in favorable anisotropic conditions in the nanofibers.

The alignment of the precursor solution strongly affects the electrospinnability of these systems [21]; therefore, the electrospinnabilities of various cellulose nanofibers were investigated using scanning electron microscopy (SEM, Figure 3a and Appendix A). Although well-developed nanofiber structures were observed for all cellulose nanofibers, the diameter distribution differed slightly (Appendix A). The diameter distribution of H-NF exhibited a standard normal distribution curve, whereas those of S-NF and P-NF exhibited bimodal and multimodal distribution curves, respectively. The average diameter and diameter deviation of the cellulose nanofibers increased with increasing cellulose content, whereas the diameter deviation increased in the order of H-NF > S-NF > P-NF. Furthermore, the analysis of N_2_ adsorption by the samples (Appendix A) demonstrated that the specific surface area slightly increased with increasing cellulose content, where H-NF exhibited the largest specific surface area of all samples. The cellulose content thus affects both the nanofiber diameter and specific surface area. Pulp-type cellulose, particularly H-pulp, exhibited improved electrospinning stability.

Based on the comprehensive analysis of the correlation between the solubility, viscosity, alignment of the precursor solutions, and the nanofiber electrospinning properties, the optimal composition ranges were plotted in ternary phase diagrams for various cellulose types (Figure 3b). The pulp-type cellulose samples exhibited a wider range of suitable manufacturing compositions than the powder-type cellulose, and H-Pulp exhibited a wider composition range than S-Pulp. Hemicellulose and lignin contents are known to attach to cellulose and reduce electrospinnability [26]. Moreover, H-Pulp possesses lower hemicellulose and lignin contents than S-Pulp and therefore exhibits superior electrospinning properties.

The mechanisms of PM removal by the fabricated cellulose nanofibers are illustrated using the SEM images obtained during PM adsorption and the results of a previous study [27] (Figure 4a). During the initial adsorption stage, a continuous PM supply begins adsorbing onto the cellulose nanofiber surface. Larger PM particles are formed subsequently by aggregation as PM migrates along the nanofibers. During the movement stage, the aggregated PM particles migrate to the nanofiber junction, creating voids where more PM can be adsorbed. As this process repeats, the new adsorption stage commences, and the newly supplied PM begins adsorbing onto the nanofiber surface. The PM removal properties of the H-NF, S-NF, and P-NF samples with different cellulose contents were investigated (Figure 4b and Table 1). The SEM images of various cellulose nanofibers after PM_2.5_ adsorption indicate that a dense layer of the largest PM was attached to the H-6.3NF surface, whereas a sparsely populated layer of the smallest PM was attached to the P-6.0NF surface. The PM adsorption properties of the H-NF, S-NF, and P-NF samples with various cellulose contents exhibited identical trends (Table 1). The PM adsorption performance of all the cellulose nanofiber samples improved with increased PM size. Although the specific surface area increased with the cellulose content (Appendix A), maximum air permeability was observed at a cellulose content of 6.3 wt%. Therefore, cellulose nanofibers with 6.3 wt% cellulose demonstrated superior PM removal efficiency and PM_2.5_ *QF* despite their low specific surface area. Furthermore, the pulp-type samples exhibited better PM removal performances than the powder-type sample owing to stronger anisotropic conditions in the precursor solutions. Among the pulp-type samples, H-NF, which possessed a larger specific surface area and better electrospinnability, demonstrated a superior PM removal performance than S-NF. The H-6.3NF system exhibited highly efficient PM removal, with a PM_2.5_ adsorption efficiency of 97.38% and PM_2.5_ *QF* of 0.28.

A schematic of toluene adsorption (as a representative VOC) on the cellulose nanofibers, which involves π–π interactions, hydrophobic effects, and van der Waals interactions [28], is depicted in Figure 5a. During adsorption, π–π interactions and hydrophobic effects are observed between toluene and the functional groups on the cellulose surface, whereas van der Waals interactions occur via toluene adsorption within the cellulose. The adsorption breakthrough curves of all samples (Figure 5b) exhibit similar shapes, indicating that they exhibit similar toluene adsorption behavior. As the cellulose content increased and the anisotropic effect enhanced, the specific surface area increased (Appendix A); thus, the breakthrough time also increased. The amount of toluene adsorbed by the samples was calculated according to these adsorption breakthrough curves (Figure 5c). All nanofiber samples exhibited increased toluene adsorption than that of the raw celluloses (H-Pulp, S-Pulp, and Powder), owing to the longer adsorption pathway in the nanofibers. In addition, the pulp-type nanofibers exhibited greater toluene adsorption than the powder-type nanofibers owing to the superior anisotropic conditions and larger specific surface areas, with H-NF and S-NF demonstrating similar amounts of toluene adsorption. These observations were confirmed through the Fourier transform-infrared (FT-IR) spectra of the H-6.3NF, S-6.3NF, and P-6.3NF samples obtained before and after toluene adsorption (Figure 5d). After toluene adsorption, the O–H peak intensity (3223 cm^−1^) increased owing to moisture adsorption from the atmosphere, whereas the C–H (2921 cm^−1^), C–C (1322 cm^−1^), and C–O–C (1030 cm^−1^) peaks also exhibited higher intensities owing to toluene adsorption [29]. In particular, the C–O–C peak intensities in the spectra of H-6.3NF and S-6.3NF were significantly higher than that of the P-6.3NF spectrum, indicating a higher degree of toluene adsorption. In addition, the amount of toluene adsorbed by H-6.0NF, H-6.3NF, and H-6.5NF increased with cellulose content, adsorbing 17.2, 18.4, and 19.8 mg/g of toluene, respectively (Figure 5c). In contrast to the PM removal process, which was influenced by the complex effects of air permeability and specific surface area, the efficiency of the toluene adsorption process increased with cellulose content owing to the increased active sites.

## 4. Conclusions

In summary, natural cellulose-based multifunctional nanofibers for the removal of PM and VOCs were manufactured using EmimAC-DMF mixed solvent systems with improved cellulose solubility and electrospinnability. The optimized cellulose precursor solution composition was achieved with a 3:7 EmimAC-DMF volume ratio, 60 min mixing time, and 6.3 wt% pulp-type celluloses. Highly efficient removal of both PM and VOCs was demonstrated, with H-6.3NF exhibiting a PM_2.5_ adsorption efficiency of 97.38%, PM_2.5_ QF of 0.28, and toluene adsorption of 18.4 mg/g. This study provides novel insights for fabricating eco-friendly multifunctional nanofiber filters for highly efficient air pollution control systems.

## Figures and Tables

**Figure 1 nanomaterials-13-01720-f001:**
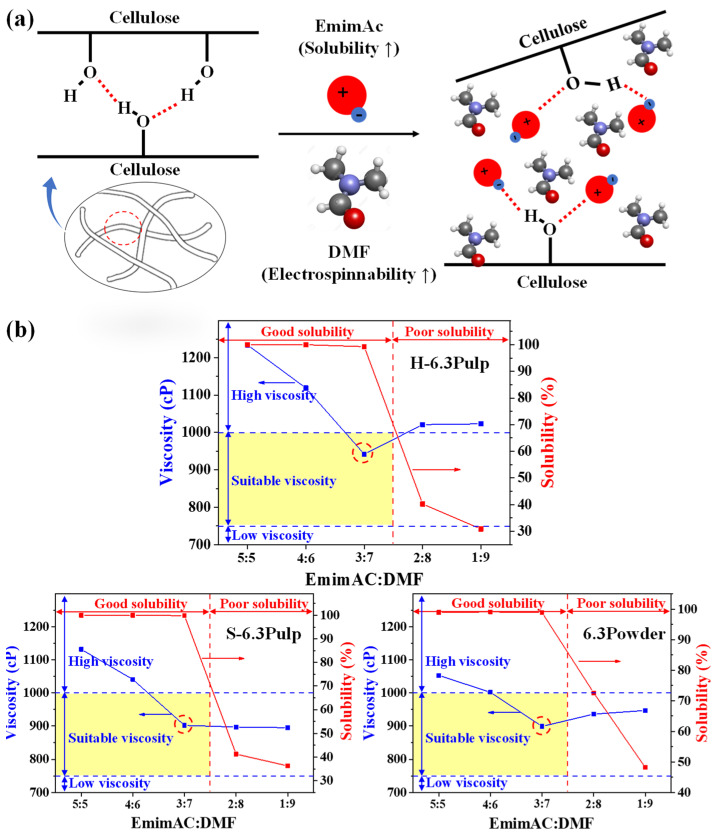
(**a**) Schematics of the cellulose dissolution mechanism in mixed solvents. (**b**) Viscosity and solubility characteristics of cellulose precursor solutions as a function of EmimAC and DMF mixing ratio.

**Figure 2 nanomaterials-13-01720-f002:**
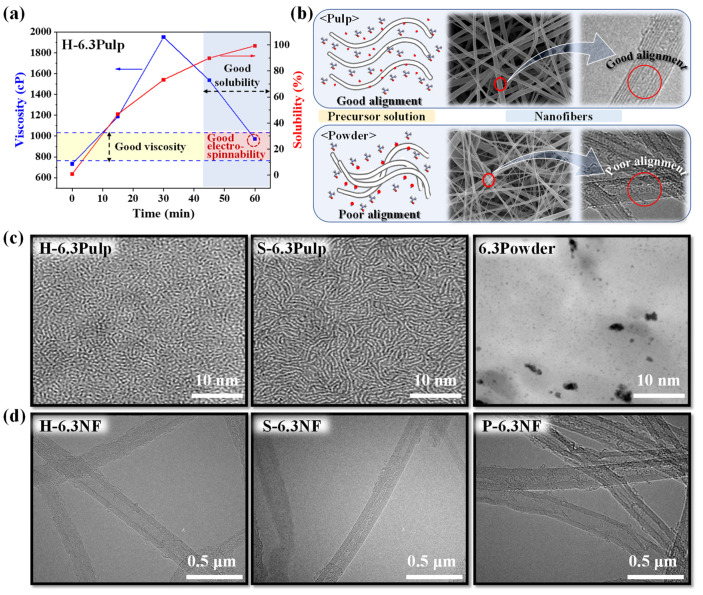
(**a**) Viscosity and solubility characteristics of H-6.3Pulp as a function of mixing time. (**b**) Schematic of the morphology conditions of the cellulose precursor solutions and cellulose nano-fibers. TEM images of (**c**) the cellulose precursor solution at a mixing time of 60 min and (**d**) the cellulose nanofibers.

**Figure 3 nanomaterials-13-01720-f003:**
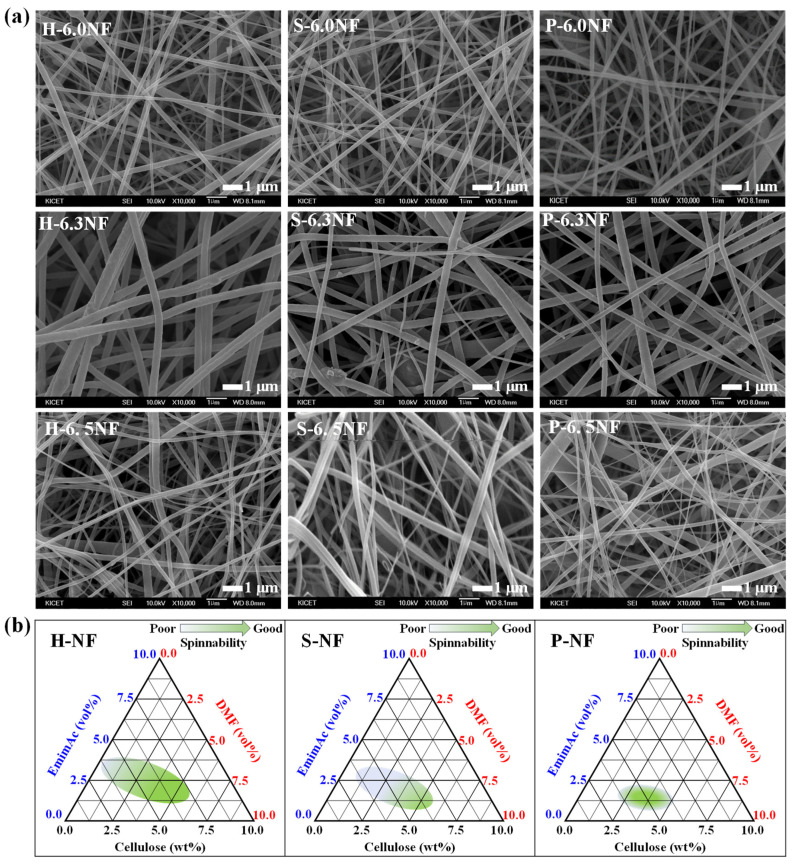
(**a**) SEM images of the H-NF, S-NF, and P-NF systems with various cellulose contents. (**b**) Ternary phase diagrams illustrating the suitable composition range for the production of the desired cellulose nanofibers.

**Figure 4 nanomaterials-13-01720-f004:**
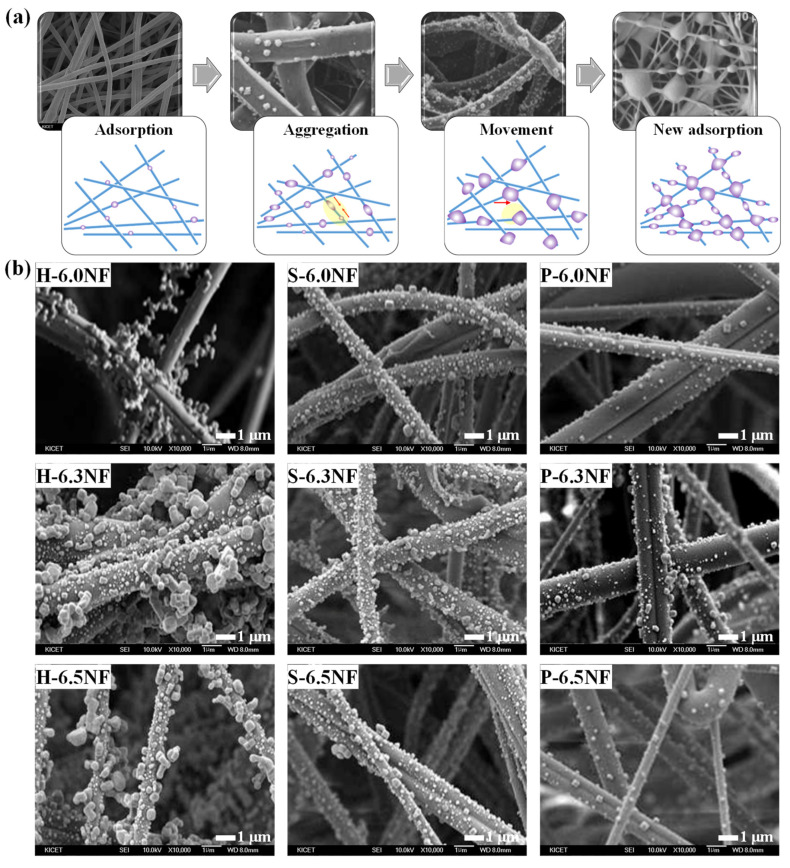
(**a**) Schematic diagrams of the PM removal mechanism by cellulose nanofibers. (**b**) SEM images of the H-NF, S-NF, and P-NF systems with various cellulose contents after PM_2.5_ adsorption.

**Figure 5 nanomaterials-13-01720-f005:**
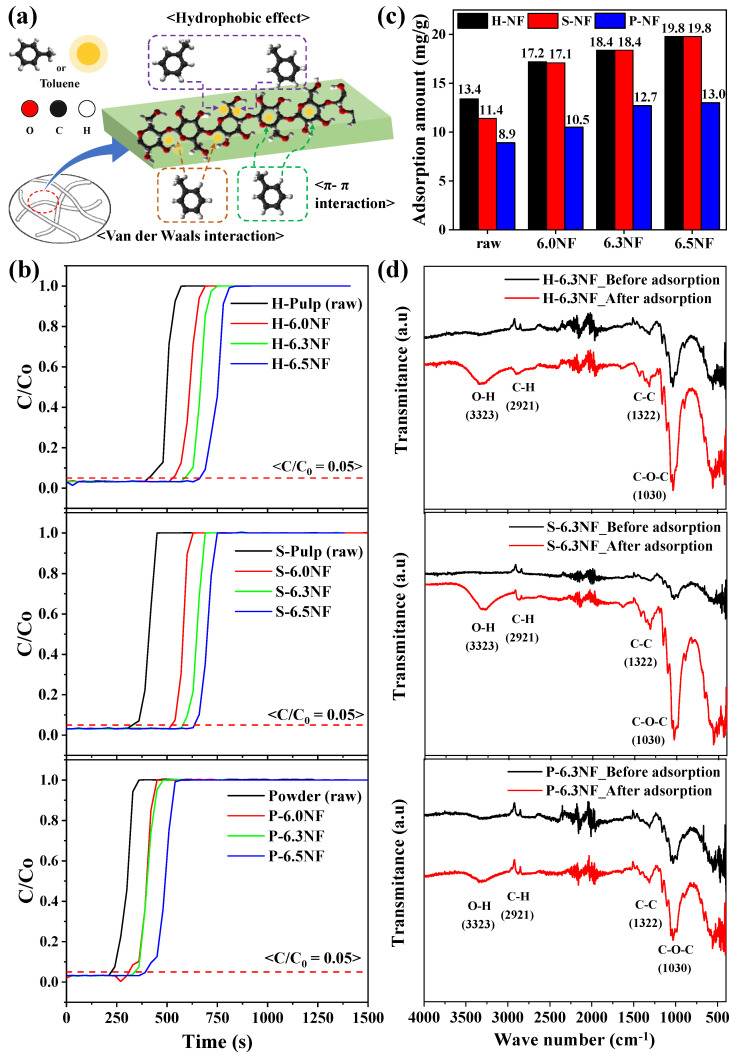
(**a**) Schematic of toluene adsorption on the cellulose nanofibers. Characteristics of toluene adsorption on the cellulose nanofibers: (**b**) adsorption breakthrough curves; (**c**) adsorption amount; and (**d**) FT-IR spectra before and after toluene adsorption.

**Table 1 nanomaterials-13-01720-t001:** PM removal properties of the H-NF, S-NF, and P-NF systems with different cellulose contents.

	Adsorption Efficiency (%)			Pressure Drop	Air Permeability	PM_2.5_ Quality Factor
	PM_0.5_	PM_1.0_	PM_2.5_	(Pa)	(cm^3^/cm^2^/s)	(Pa^–1^)
H-6.0NF	85.36	88.14	89.12	98	277	0.24
H-6.3NF	92.25	97.25	97.38	105	284	0.28
H-6.5NF	88.25	93.22	95.25	97	270	0.23
S-6.0NF	83.24	87.15	92.36	96	274	0.22
S-6.3NF	86.25	88.83	94.21	102	283	0.27
S-6.5NF	83.12	89.12	95.31	93	269	0.21
P-6.0NF	74.15	76.81	79.17	88	281	0.19
P-6.3NF	75.43	77.22	81.21	99	277	0.25
P-6.5NF	73.25	76.22	80.05	86	264	0.18

## Data Availability

Data presented in this study are available by requesting from the corresponding author.

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
