# Peer review of "Natural Cellulose-Based Multifunctional Nanofibers for the Effective Removal of Particulate Matter and Volatile Organic Compounds"

_nanomaterials, 2023, doi:10.3390/nano13111720_

Round 1

Reviewer 1 Report

The submitted manuscript entitled "Natural cellulose-based multifunctional nanofibers for the effective removal of particulate matter and volatile organic compounds " performed by Sang Hyun Ji and Ji Sun Yun deals with utilization of cellulose based electrospun nanofibrous materials for removal both of particulate matter and VOCs. Currently, the utilization of nanofibrous mats prepared by electrospinning technique in various areas involving membrane separation, filtration, catalysis or biomedical applications represents a valuable task. The optimization of cellulose electrospun solution is very well described in the manuscript. However, a part related to textural analysis of the prepared electrospun materials should be clarify and explain. Authors stated in the Supplementary materials that prepared electrospun materials revealed quite huge specific surface area evaluated by B.E.T. method (around 1000 m2/g in all cases). This is quite unusual, as follows from the literature, typical specific surface area of electrospun materials corresponds to interval 10 to 20 m2g. Even for nanofibers with internal porosity, aforementioned textural characteristic corresponds to about 100 m2/g. Another discrepancy is related to the pore volume. Authors given in Table S2 values ranging from 0.04 to 0.11 cm3/g. However, these values seem to be very low compared to the corresponding specific surface area. Authors should give corresponding nitrogen adsorption isotherms measured and also (micro)pore-size distribution functions with appropriate discussion.

Moderate editing of English language.

Author Response

Thank you for your kind comments. The specific surface area of H-Pulp (raw), S-Pulp (raw) and Powder (raw) were 936 m2/g, 848 m2/g and 716 m2/g, respectively. The specific surface area of the H-NF, S-NF, and P-NF samples with various cellulose contents were overall improved than each raw materials, due to the well-formed nanofiber structures. This tendency was also reported in our previous work (Materials & Design 229 (2023) 111926). However, according to your comments, I added the nitrogen sorption isotherms and pore-size distribution curves of all samples in Figure S3, and further discussed as follows:

Reviewer 2 Report

The current work by Sang and co-workers is very interesting, with the soundness of the results well presented. Thus, I do recommend publication after minor considerations, as highlighted below:

1. How reusable were the fibres post the adsorption?

2. Can authors provide elemental mapping and EDS of the SEM images after adsorption so as to ascertain only toluene was removed.

3. What was the selectivity of the fibres to toluene in the presence of other VOCs? This data must be presented to ensure the efficiency of these filters.

Minor grammatical errors should be looked at in detail.

Author Response

  1. How reusable were the fibres post the adsorption?

=> Thank you for your kind comments. According to your comments, the degree of reuse of nanofibers is one of the key issues in the field of air purifiers, and to confirm this degree, not only adsorption studies but also desorption studies of pollutants should be conducted. VOCs were desorbed while raising the temperature, and a decrease in adsorption properties of about 5% was observed when adsorption and desorption were performed about 10 times. However, in the case of PM filters, research on filter regeneration is still in initial stage, and there is no standardized method, and rather, the issue of filter waste treatment is more important. In this regard, the development of biodegradable cellulose nanofibers is required. However, since the reusability of filters is an important issue, we will further study the reusability of filters of the cellulose nanofibers in the follow-up research.

  1. Can authors provide elemental mapping and EDS of the SEM images after adsorption so as to ascertain only toluene was removed.

=> Thank you for your kind comments. The results of elemental mapping and EDS of the SEM images after adsorption showed the content of elements such as C, O and H. However, since the elements of toluene and cellulose are almost the same, it is very difficult to distinguish the unique elements only in toluene after adsorption. Toluene adsorption was determined using a catalytic reaction system. The adsorption isotherms of 1000 ppm toluene based on carrier gas of Ar (inert gas) were measured at 35 °C with a flow rate of 50 cm3/min, whereas the toluene concentration in the outlet flow was recorded using gas chromatography. In other words, since only toluene was inflowed excepting inert gas, and toluene peak was analyzed by GC, it is believed that toluene was removed. Furthermore, the FT-IR spectra before and after toluene adsorption in Figure 5(d) showed that a degree of toluene adsorption according to the different types of the cellulose nanofibers.

  1. What was the selectivity of the fibres to toluene in the presence of other VOCs? This data must be presented to ensure the efficiency of these filters.

=> Thank you for your kind comments. Since most VOCs are adsorbed on similar adsorption sites, cellulose nanofibers will not selectively adsorb only toluene. For selective adsorption of toluene, it is considered necessary to provide functional groups to the surface of cellulose nanofibers. According to your comments, we will further study Cellulose nanofiber surface functionalization in the follow-up research.

Reviewer 3 Report

Major weaknesses in the manuscript's rationality contribute to its incompleteness and lack of scientific rigor (ID nanomaterials-2397472). Thus, a major revision is necessary for the following reasons:

1. It is recommended to modify the abstract to supplement the significance and value of the research, and to propose the focus and innovation of the research.

2. It is necessary to supplement the method of solubility, polarized light photos of solution in the dissolution process to present the dissolution process more specifically.

3. Regarding the description "precursor solution anisotropy": I think it is completely wrong and comical.

First, ionic liquids are common solvents for dissolving cellulose, making cellulose raw materials into countless, disordered cellulose molecular chains in solution. The nanofibers observed in the electrospinning process are essentially the process of cellulose molecular chain aggregation, that is, cellulose regeneration.

Second, the presented AFM and TEM data of the cellulose solution in Figures 2c and 2d are extremely misleading. Concentrated cellulose solutions cannot be characterized by atomic force, and usually need to be diluted by more than 10,000 times. Obviously, the presented AFM data by the authors does not show the cellulose molecular chain, so a higher magnified AFM image needs to be provided, otherwise the surface cellulose will not dissolve in the solution (only dispersed as nanofibers). In addition, it is recommended to also take a TEM of the diluted cellulose solution to show the dissolution behavior of the cellulose.

4. Considering that the structure of regenerated cellulose nanofibers is significantly related to the adsorption effect, the size distribution, crystallinity, surface charge density of nanofibers needs to be provided. In addition, the author should re-summarize and verify the mechanism of PM adsorption by nanocellulose, which should at least be related to one of the five adsorption mechanisms mentioned in the Introduction.

5. The quality of the pictures need to be improved, including supplementary scales, unified label positions, and improved viewing of schematic diagrams.

6. Please design the experiment of simultaneously capturing PM and VOCs with nanocellulose to verify the feasibility in complex application environments and prove whether the adsorption sites influence each other. In addition, please add a comparison chart of the adsorption value of nanocellulose for PM and VOCs with the performance of commercial products and other commonly used materials to demonstrate the practical application value of cellulose nanofiber materials.

7. Check the references.

Author Response

  1. It is recommended to modify the abstract to supplement the significance and value of the research, and to propose the focus and innovation of the research.

=> Thank you for your kind comments. According to your comments, I modified the abstract as follows:

  1. It is necessary to supplement the method of solubility, polarized light photos of solution in the dissolution process to present the dissolution process more specifically.

=> Thank you for your kind comments. There are various solubility measurement methods using various tools such as UV spectrophotometry, Raman spectroscopy, and a solubility measurement equipment. In this work, quantitative solubility values according to the mixing process were measured using a solubility measurement equipment (NEPHELOster, BMG Labtrch, Germany). However, I think your opinion on the solubility analysis method through polarized light photography is reasonable, and according to your comments, we will further study the imaged solubility analysis of the cellulose precursor solution in the follow-up research.

  1. Regarding the description "precursor solution anisotropy": I think it is completely wrong and comical.

First, ionic liquids are common solvents for dissolving cellulose, making cellulose raw materials into countless, disordered cellulose molecular chains in solution. The nanofibers observed in the electrospinning process are essentially the process of cellulose molecular chain aggregation, that is, cellulose regeneration.

Second, the presented AFM and TEM data of the cellulose solution in Figures 2c and 2d are extremely misleading. Concentrated cellulose solutions cannot be characterized by atomic force, and usually need to be diluted by more than 10,000 times. Obviously, the presented AFM data by the authors does not show the cellulose molecular chain, so a higher magnified AFM image needs to be provided, otherwise the surface cellulose will not dissolve in the solution (only dispersed as nanofibers). In addition, it is recommended to also take a TEM of the diluted cellulose solution to show the dissolution behavior of the cellulose.

=> Thank you for your kind comments. Figures 2c and 2d are the AFM and (d) TEM images of the cellulose precursor solution, not cellulose nanofibers. These figures were measured in the cellulose precursor solution before nanofiber formation, not after electrospinning process, and it is believed that the cellulose, not the nanofibers, are dispersed in the mixed solvents. Furthermore, in our previous work (Materials & Design 229 (2023) 111926), flow-induced structural anisotropy characteristics of the cellulose precursor solution were also analyzed by rheological behaviors, AFM, and TEM. We also analyzed AFM and TEM using dilute solutions. However, according to your comments, we will further study of AFM and TEM analysis according to the degree of dilution in the follow-up research.

  1. Considering that the structure of regenerated cellulose nanofibers is significantly related to the adsorption effect, the size distribution, crystallinity, surface charge density of nanofibers needs to be provided. In addition, the author should re-summarize and verify the mechanism of PM adsorption by nanocellulose, which should at least be related to one of the five adsorption mechanisms mentioned in the Introduction.

=> Thank you for your kind comments. In the Introduction, I mentioned the commercialized PM removal methods such as sieving, gravity settling, inertial impaction, interception, diffu-sion, and electrostatic attraction. However, because these commercial filters comprise synthetic polymers, secondary pollutants are generated after use, and compact and slim air filters constructed from eco-friendly natural materials are in demand. In this regard, we suggested the cellulose nanofibers as a new indoor air cleaning filter material in this work. The PM removal principal of nanofibers was reported that the PM particles wrap around the nanofiber, migrate along the nanofiber, and aggregate to form larger particles (Nature Communications 6 (2015) 6205, Materials & Design 229 (2023) 11192). Based on these previous studies (Nature Communications 6 (2015) 6205, Materials & Design 229 (2023) 11192) and the results of SEM analysis at each stage of PM adsorption, we plotted schematic diagrams of the PM removal mechanism by cellulose nanofibers in Figure 4(a). Furthermore, we re-summarized mechanism the in the paper as follow:

“During the initial adsorption stage, a continuous PM supply begins adsorbing onto the cellulose nanofiber surface. Larger PM particles are formed subsequently by aggregation as PM migrates along the nanofibers. During the movement stage, the aggregated PM particles migrate to the nanofiber junction, creating voids where more PM can be adsorbed. As this process repeats, the new adsorption stage commences, and the newly supplied PM begins adsorbing onto the nanofiber surface.”

However, according to your comments, we will further study of the PM removal mechanism in different environments in the follow-up research.

  1. The quality of the pictures need to be improved, including supplementary scales, unified label positions, and improved viewing of schematic diagrams.

=> Thank you for your kind comments. According to your comments, I overall improved the quality of the pictures.

  1. Please design the experiment of simultaneously capturing PM and VOCs with nanocellulose to verify the feasibility in complex application environments and prove whether the adsorption sites influence each other. In addition, please add a comparison chart of the adsorption value of nanocellulose for PM and VOCs with the performance of commercial products and other commonly used materials to demonstrate the practical application value of cellulose nanofiber materials.

=> Thank you for your kind comments. In this work, we demonstrated that the cellulose nanofibers have both PM and VOC removal capabilities. Based on the possibility we suggested, we will further study of the adsorption properties for each pollutant in the simulated indoor environment in which PM and VOCs are mixed in a follow-up study. In addition, based on the development of evaluation technology in the simulated environment, it will be possible to compare the properties of the commercial products and developed products. Please understand that the experimental results in an environment that simulates the real environment have been carried out through follow-up research.

  1. Check the references.

=> Thank you for your kind comments. According to your comments, I checked the references, and revised them as appropriate.

Round 2

Reviewer 1 Report

I recommend to accept in present form.

Author Response

Thank you for your kind comments. It is an honor for me to have the opportunity of our manuscript published in Nanomatierials.

Reviewer 3 Report

The authors avoided almost all revisions proposed by the reviewers regarding experimental additions and principled conceptual discussions (nanomaterials-2397472-Revised Version). Therefore, in this round of revisions, if the author still cannot provide convincing data, I will directly reject the manuscript.

1. The polarized light photos of solution in the dissolution process must be supplemented.

2. The description "precursor solution anisotropy" is completely wrong and comical. The author must delete this description in the article. The author claimed that “Figures 2c and 2d are the AFM and (d) TEM images of the cellulose precursor solution, not cellulose nanofibers. These figures were measured in the cellulose precursor solution before nanofiber formation, not after electrospinning process, and it is believed that the cellulose, not the nanofibers, are dispersed in the mixed solvents”. First of all, the magnification of AFM data is obviously low, resulting in the measured size and height of the so-called cellulose molecular chains significantly deviating from the actual molecular chains. Second, the AFM data also do not agree with the TEM data. The TEM does not exhibit any distributed anisotropic behavior. Therefore, the authors had to make textual revisions, including 1) deleting any description of solution anisotropy; 2) compensating for detailed descriptions of the AFM and TEM sample preparation process; 3) deleting AFM data or replacing it with a picture of the 10nm scale range

Author Response

The authors avoided almost all revisions proposed by the reviewers regarding experimental additions and principled conceptual discussions (nanomaterials-2397472-Revised Version). Therefore, in this round of revisions, if the author still cannot provide convincing data, I will directly reject the manuscript.

  1. The polarized light photos of solution in the dissolution process must be supplemented.

=> Thank you for your kind comments. I think the analysis result you want is like the picture below. 

I totally agree with you opinion that among the various methods for measuring the solubility of cellulose, polarized light photography can be a valuable tool for studying the dissolution process in crystalline systems, providing insights into crystal formation, morphology, and dissolution rates. However, its applicability may be limited to certain systems, and the technique requires specialized equipment and expertise for accurate interpretation. Please understand that it is impossible to provide an upload period as short as 5 days, as it takes a lot of time and effort to accurately analyze the analysis. In this regard, we will further study the polarized light photo analysis of the cellulose precursor solution in the follow-up research.

  1. The description "precursor solution anisotropy" is completely wrong and comical. The author must delete this description in the article. The author claimed that “Figures 2c and 2d are the AFM and (d) TEM images of the cellulose precursor solution, not cellulose nanofibers. These figures were measured in the cellulose precursor solution before nanofiber formation, not after electrospinning process, and it is believed that the cellulose, not the nanofibers, are dispersed in the mixed solvents”. First of all, the magnification of AFM data is obviously low, resulting in the measured size and height of the so-called cellulose molecular chains significantly deviating from the actual molecular chains. Second, the AFM data also do not agree with the TEM data. The TEM does not exhibit any distributed anisotropic behavior. Therefore, the authors had to make textual revisions, including 1) deleting any description of solution anisotropy; 2) compensating for detailed descriptions of the AFM and TEM sample preparation process; 3) deleting AFM data or replacing it with a picture of the 10nm scale range

=> Thank you for your kind comments. According to your comments, I deleted AFM images and any description of solution anisotropy. Furthermore, I detailly described the TEM sample preparation process. I revised our manuscript as follows: 

Round 3

Reviewer 3 Report

No other questions.